# Effect of Tannin Inclusion on the Enhancement of Rumen Undegradable Protein of Different Protein Sources

Kalista E. Loregian [1,2], David A. B. Pereira [1], Fernanda Rigon [2], Elaine Magnani [1], Marcos I. Marcondes [3], Eduardo A. Baumel [2], Renata H. Branco [1], Pedro Del Bianco Benedeti [2] and Eduardo M. Paula [1,*]

1   Centro APTA Bovinos de Corte, Instituto de Zootecnia, Sertãozinho 14160-970, SP, Brazil; kloregian71@gmail.com (K.E.L.); davidzootec@gmail.com (D.A.B.P.); lainemag@hotmail.com (E.M.); rhbarnandes@gmail.com (R.H.B.)
2   Department of Animal Sciences, Universidade do Estado de Santa Catarina, Chapecó 89815-630, SC, Brazil; fernanda.rigon@unesp.br (F.R.); edu.abaumel@gmail.com (E.A.B.); pedro.benedeti@udesc.br (P.D.B.B.)
3   Department of Animal Sciences, Washington State University, Pullman, WA 99164, USA; marcos.marcondes@wsu.edu
*   Correspondence: emarostegandepaula@gmail.com; Tel.: +55-(16)-3475-9162

**Abstract:** Tannins can be utilized to increase rumen undegradable protein (RUP) by their capacity to form complexes with diverse nutrients present in the feed. In that regard, high-performance ruminants demand elevated RUP levels. The objective of this study was to evaluate the effects of incorporating varying levels of tannin into three protein sources (cottonseed, peanut, and soybean meals) on ruminal kinetic parameters, ruminal fermentation, and intestinal digestibility. Thus, three in situ experiments were conducted to investigate the ruminal degradation kinetics, where Fraction A represents the soluble portion, Fraction B relates to the portion potentially degraded in the rumen, and kd denotes the degradation rate of Fraction B, for both dry matter (DM) and crude protein (CP) in the rumen. Additionally, the study assessed dry matter effective degradability (ED), rumen undegradable protein (RUP), and intestinal digestibility (ID). These experiments utilized three cannulated animals for the in situ incubations. Regarding cottonseed meal in terms of DM degradation kinetics, tannin inclusion had a quadratic effect on fraction A ($p < 0.01$), B ($p = 0.10$, trend), kd ($p = 0.03$), and ED ($p < 0.01$). Fraction A of CP had a cubic effect ($p = 0.03$), being greater for the control compared with the other treatments. The inclusion of tannin linearly increased RUP ($p < 0.01$). The RUP proportion increased 29, 33, and 45% when 20, 40, and 60 g/kg tannin were used, respectively, compared to the control. For peanut meal, the A fraction of protein and RUP responded quadratically as tannin was included in peanut meal ($p < 0.01$). However, tannin levels did not affect fraction B of protein and ID. Regarding soybean meal, fractions A and B of DM and ED had cubic effects ($p < 0.01$), being greater for the control compared with the other treatments, and responded quadratically as tannin increased. Also, tannin inclusion had a cubic effect on fractions A and B of protein, RUP, and ID ($p < 0.01$). The cubic behavior showed greater B fraction and ID and lower A fraction and RUP for the control compared other treatments ($p < 0.01$). Tannins offer a promising avenue for elevating RUP levels in diets featuring cottonseed and peanut meals. Nevertheless, no advantages were observed when treating soybean meal with tannin.

**Keywords:** beef cattle; ruminal degradation; protein degradation; tannins

## 1. Introduction

Ruminant animals still exhibit significantly lower feed efficiency compared to poultry and swine, indicating the pressing need for further advancements in this area [1–3]. Furthermore, protein is the costliest component among the feed constituents [4]. Hence, there is a need for research endeavors to optimize the utilization of this nutrient, which in the case of ruminants is classified into rumen-degradable protein (RDP) and rumen-undegradable

protein (RUP) [5]. The RDP undergoes conversion into amino acids by rumen microorganisms, thereby serving as the primary source of metabolizable protein for ruminants [4]. However, microbial protein alone is insufficient to meet the requirements of high-producing animals [6]. Thus, high-producing animals require a greater RUP contribution to fulfill their metabolizable protein needs.

Soybean, peanut, and cottonseed meals are the main protein sources used in feedlot cattle production in Brazil [7]. Although widely utilized, a significant portion of these feeds' protein is fermented in the rumen [8,9]. Thus, researchers have sought processing techniques (physical and chemical) to enhance protein protection against ruminal fermentation and improve protein utilization [10]. Several processing methods have been developed to manipulate the ruminal degradation of protein sources with the objective to increase feedstuff RUP, including autoclaving [11], microwave treatment [12,13], toasting or irradiation [14], heating [15,16], malic acid [16], xylose [17], and tannins [18,19]. Among these processing techniques, the addition of condensed tannins is noteworthy. Condensed tannins, recognized for their anti-nutritional properties, can form stable complexes with protein when added at low concentrations, resulting in increased resistance to rumen microbial degradation [20,21]. Thus, studies have shown that the addition of condensed tannins, at low concentrations, can increase RUP [22]. However, the optimal levels of tannin utilization for protein protection are not yet well-established. It is important to note that this protein protection should not interfere with intestinal protein digestibility, should not cause harm to animals, and should not adversely affect the ruminal microbial population [23].

There are uncertainties regarding the use of tannins to increase RUP, such as the lack of specific dosages and their effects on different feed ingredients, such as peanut and cottonseed meals. Furthermore, there are limited studies on the effectiveness of increasing RUP supply on ruminal fermentation. In this regard, the use of in vitro techniques allows for controlled studies and provides relevant information on ruminal fermentation of feedstuffs [24,25]. Therefore, this study aimed to evaluate different levels of tannin inclusion in three protein sources (cottonseed, peanut, and soybean meals) on ruminal kinetic parameters, ruminal fermentation, and intestinal digestibility. Our hypothesis is that tannin treatment can effectively protect the protein in these feed ingredients against ruminal fermentation, leading to an increase in RUP and improving nutrient utilization efficiency.

## 2. Materials and Methods

### 2.1. Location and Ethical Approval

The experiments took place at the Beef Cattle Research Center, located at the Institute of Animal Science in Sertãozinho, São Paulo, Brazil. This study was conducted in full compliance with the guidelines set forth by the Animal Use Ethics Committee of the Institute of Animal Science. The committee approved the protocol for animal care and handling under the reference number 249-19.

### 2.2. Experimental Designs and Chemical Analysis

Three experiments were performed for individual feed evaluation: cottonseed (Exp.1), peanut (Exp.2), and soybean (Exp.3) meals. Within each experiment, feed were submitted to four levels of tannin (85% condensed and 15% hydrolyzed from *Acacia mearnsii*, Tanac SA, Montenegro, RS, Brazil) including: 0, 20, 40, and 60 g/kg (DM basis). Furthermore, a commercial soybean-based product was also evaluated as positive control (SoyPass®, Nutron Cargill, São Paulo, SP, Brazil) in Exp.3.

For in situ trials, three Nellore bulls (BW of 397 ± 51 kg, 24 months old, and a body condition score of 3.5), cannulated in the rumen, were used for ingredients incubation. The experimental design included a randomized complete block (animal). For each ingredient, simultaneously in each animal, bags were incubated in the rumen for 0, 2, 4, 8, 12, 24, or 48 h. Filter bags were incubated for each treatment in duplicate in each animal and timepoint, totaling 42 observations per treatment. In the in vitro trials, a setup comprising four 4-L digestion vessels (TE-150, Tecnal Equipamentos Científicos, Piracicaba, SP, Brazil) was employed. These vessels were equipped with a gentle rotational mechanism and a temperature controller and used in a 24 h fermentation batch. Thus, the experimental design included a randomized complete block (vessel), with a total of 26 filter bags (experimental units) incubated for each treatment.

The ingredients used in these studies underwent grinding using a 2 mm screen (Wiley mill; Thomson Scientific Inc., Philadelphia, PA, USA) for all incubations and analyses. Subsequently, the samples were subject to chemical analyses, including dry matter (DM; method G-003/1), ash (method M-001/1), crude protein (method N-001/1), and ether extract (method G-005/1), following the procedures outlined in Detmann et al. [26]. The organic matter (OM) content was calculated as the difference between DM and ash. To determine neutral detergent fiber, the samples were treated with alpha thermo-stable amylase, with the omission of sodium sulfite, in accordance with the method by Van Soest et al. [27] and adapted for the Ankom200 Fiber Analyzer (Ankom Technology, Macedon, NY, USA). Detailed information on the chemical compositions of the experimental ingredients can be found in Table 1. The tannin treatments were prepared as follows: 600 g of each sample was weighed using a digital scale (Mettler Toledo, model ME2002E, Polaris Parkway, Columbus, OH, USA) and placed in metal molds. Then, the tannin was weighed and homogenized with each sample. After homogenization, distilled water (in a 1:2 ratio) was added to the mixture, which was left to stand for 6 h. After resting, the sample was transferred to a ventilated oven at 60 °C for 72 h.

**Table 1.** Chemical composition of protein sources.

| Composition | Cottonseed Meal | Peanut Meal | Soybean Meal |
|---|---|---|---|
| Dry matter, g/kg | 910 | 915 | 910 |
| Crude protein, g/kg DM | 543 | 628 | 524 |
| Ether extract, g/kg DM | 16.4 | 12.8 | 19.4 |
| Neutral detergent fiber, g/kg DM | 174 | 164 | 157 |
| Ash, g/kg DM | 63.5 | 53.9 | 62.3 |

*2.3. In Situ Procedures and Calculations*

The animals were situated in an enclosed barn and placed in individual tie stalls. They were provided with a diet consisting of 40% forage and 60% concentrate, which included a mixture of 60% corn silage, 24.9% dry ground corn, 13% soybean meal, 0.2% urea, and 1.9% mineral supplement. Bulls were adapted to this diet for 14 d before the commencement of the study, and they had continuous access to water. Each ingredient was weighed and then inserted into Nylon bags (Sefar Nitex, Switzerland, Fairport, NY, USA) with a porosity of 50 µm and a surface area of 400 cm$^2$. These bags were subsequently placed inside each animal for incubation, ensuring a bag surface area to mass ratio of 15 mg/cm$^2$. Samples were incubated in the rumen by attaching the bags to a steel chain with a weight at the end to allow for continual immersion within ruminal contents. Bags were placed into the rumen in the reverse order of incubation hours so that all bags were removed at the same time for washing.

After removal, the bags were immersed in an ice-cold saline solution for 15 min to halt microbial activity and detach bacteria from the feed fraction. Subsequently, the bags underwent a thorough washing in a washing machine using cold running tap water until the rinsing water became clear. Bags designated as "0 h" were not subjected to rumen incubation but were rinsed alongside the incubated bags. Following this, the bags were dried in an oven at 55 °C for 72 h. Upon completion of the drying process, each bag was individually weighed. The residues from each diet were carefully extracted from the bags and placed into labeled plastic bags to create a sample for each diet corresponding to each animal and incubation time. These residual samples from different time points in the bags were utilized to estimate the parameters of ruminal degradation.

The DM and CP degradation profiles were estimated asymptotic function [28]:

$$Yt = A + B \times (1 - e - (kd \times t)),$$

where Yt is the fraction degraded in time 't', g/kg; A is the water-soluble fraction, g/kg; B is the potentially degradable water-insoluble fraction, g/kg; kd is the degradation rate of fraction b, $h^{-1}$; and t is time, h.

The effective degradability (ED, g/kg) of DM was calculated using the model [29]:

$$ED = A + [B \times kd/(kd + kp) \times e\text{-}kpt),$$

where A is the water-soluble fraction, g/kg; B is the potentially degradable water-insoluble fraction, g/kg; kd is the degradation rate of fraction b, $h - 1$; t is time, h; and kp is the rumen passage rate (k) of 0.074 $h^{-1}$, obtained from the equation developed for concentrates [30].

The rumen undegradable protein (RUP) content in ingredients was calculated as:

$$RUP = B \times [kp/(kp + kd)],$$

where B is the potentially degradable water-insoluble fraction, g/kg; kd is the degradation rate, $h^{-1}$; kp = passage rate, $h^{-1}$.

### 2.4. Intestinal Digestibility Procedures and Calculations

In the context of in vitro trials, a setup was utilized that included four digestion vessels, each with a capacity of 4 L (TE-150, Tecnal Equipamentos Científicos, Piracicaba, SP, Brazil). These vessels were equipped with a controlled system to ensure slow rotation and maintained temperature, enabling a 24 h fermentation batch. To assess the intestinal digestion of RUP, a three-step in vitro procedure was employed [31,32]. In brief, 0.335 g (on a dry matter basis) from the 12 h timepoint of the in situ incubation for each ingredient was weighed and placed into R520 bags (Ankom Technology, Macedon, NY, USA). These bags were then sequentially incubated with constant rotation at 39 °C using a TE-150 incubator (Tecnal Equipamentos Científicos, Piracicaba, SP, Brazil). The incubation process involved the use of pepsin solution (P-7000, Sigma, St. Louis, MO, USA) for 1 h, followed by pancreatin solution (P-7545, Sigma, St Louis, MO, USA) for 24 h. Upon completion of the incubation period, the bags were rinsed with tap water until the effluent water became clear. Subsequently, the samples were dried in an oven at 60 °C for 48 h. The residues remaining in the bags were then analyzed to determine the dry matter (DM) and nitrogen (N) content.

### 2.5. Statistical Analysis

The DM and CP fractions, ED, RUP, and ID were first determined for each replication and compared using a completely randomized model design. The parameters were compared through a regression analysis, with differences considered statistically significant at

a level of $p \leq 0.05$ and trending when $0.05 < p \leq 0.10$. Thus, the levels of tannin inclusion were examined for linear, quadratic, and cubic responses using the following model:

$$Y_{ijk} = B_0 + B_1 X_i + B_2 X_i^2 + P_j + A_k + e_{ijk}$$

where $Y_{ijk}$ represents the response variable obtained from the $i$th level of tannin in the diet of the $j$th incubation and the $k$th experimental unit. The indices $i$, $j$, and $k$ denote the levels of inclusion of tannin, the random factor of incubation, and the random factor of bottle, respectively. The regression parameters of the model are denoted as $B_0$, $B_1$, and $B_2$. The $X_i$ represents the effect of the $i$th level of the fixed quantitative factor (inclusion of tannin), $P_j$ represents the effect of the level of the random factor incubation, $A_k$ represents the effect of the level of the random factor, and $e^{ijkl}$ represents the residual error, assumed to follow a normal distribution $(0, s^2)$. All analyses were run using the PROC GLIMMIX of SAS (SAS on Demand, online version).

## 3. Results

### 3.1. Cottonseed Meal

Ruminal degradation parameters of DM and CP, as well as protein ID can be found in Table 2 and Figure 1. Regarding DM degradation kinetics, tannin inclusion had linear and quadratic effects on fraction A ($P_{Lin.} < 0.01$; $P_{Quad.} < 0.01$), B ($P_{Quad.} = 0.10$, trend), kd ($P_{Lin.} = 0.10$, trend; $P_{Quad.} = 0.03$), and ED ($P_{Quad.} < 0.01$). Fraction A of CP had linear and cubic effects ($P_{Lin.} < 0.01$; $P_{Cub.} = 0.03$), being greater for the control compared with the other treatments and responded quadratically as tannin increased. Tannin levels did not affect Fraction B of CP ($P_{Lin.} = 0.72$; $P_{Quad.} = 0.77$; $P_{Cub.} = 0.87$). The inclusion of tannin linearly increased RUP ($p < 0.01$). Compared to the control (0 g/kg tannin), the RUP proportion increased 29, 33, and 45% when 20, 40, and 60 g/kg tannin were used, respectively. Furthermore, ID had a cubic effect ($p < 0.01$), being greater for the control compared with the other treatments and responded quadratically as tannin increased.

**Table 2.** Effects of tannin inclusion on rumen degradation parameters of cottonseed meal.

| Item [1] | Tannin Inclusion, g/kg | | | | SEM [2] | *p*-Value | | |
|---|---|---|---|---|---|---|---|---|
| | 0 | 20 | 40 | 60 | | Linear | Quadratic | Cubic |
| Dry matter | | | | | | | | |
| A, g/kg | 194 | 113 | 104 | 112 | 34.7 | <0.01 | <0.01 | 0.13 |
| B, g/kg | 595 | 629 | 644 | 599 | 7.93 | 0.82 | 0.10 | 0.71 |
| kd, h$^{-1}$ | 0.059 | 0.049 | 0.041 | 0.051 | 0.002 | 0.10 | 0.03 | 0.45 |
| Crude protein | | | | | | | | |
| A, g/kg | 345 | 315 | 331 | 293 | 5.76 | <0.01 | 0.71 | 0.03 |
| B, g/kg | 581 | 572 | 561 | 569 | 10.4 | 0.72 | 0.77 | 0.87 |
| kd, h$^{-1}$ | 0.084 | 0.060 | 0.046 | 0.044 | 0.001 | <0.01 | 0.08 | 0.92 |

[1] A, water-soluble fraction; B, potentially degradable water-insoluble fraction; kd, degradation rate of fraction B; [2] SEM, standard error of the mean.

### 3.2. Peanut Meal

The effects of tannin treatment in peanut meal can be found in Table 3 and Figure 2. For DM kinetics, tannin inclusion had a quadratic effect on fraction A ($p < 0.01$). Furthermore, fraction B had a cubic effect ($p < 0.01$), being lower for the control compared with the other treatments and responded quadratically as tannin increased. On the other hand, tannin levels did not affect kd ($P_{Lin.} = 0.56$; $P_{Quad.} = 0.38$; $P_{Cub.} = 0.15$). Nevertheless, the inclusion of tannin linearly decreased ED ($p = 0.02$). Regarding CP degradation kinetics, tannin levels did not affect fraction B ($P_{Lin.} = 0.08$; $P_{Quad.} = 0.08$; $P_{Cub.} = 0.29$) and ID ($P_{Lin.} = 0.37$; $P_{Quad.} = 0.70$; $P_{Cub.} = 0.84$). The A fraction ($p < 0.01$) and RUP ($p = 0.05$) responded quadratically as tannin was included in peanut meal.

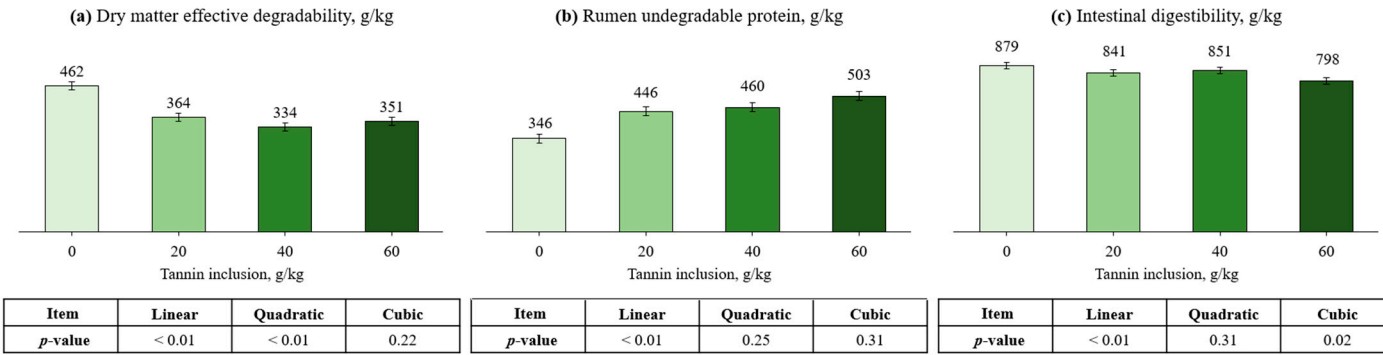

**Figure 1.** Effects of tannin inclusion on dry matter effective degradability ((**a**), ED), rumen undegradable protein ((**b**), RUP), and intestinal digestibility (**c**), of cottonseed meal. ED = A + [B × kd/(kd + kp) × e-kt] [29]; RUP = B × [kp/(kp + kd)], where A is the water-soluble fraction, g/kg; B is the potentially degradable water-insoluble fraction, g/kg; kd is the degradation rate of fraction b, h$^{-1}$; t is time, h; and kp is the rumen passage rate of 0.074 h$^{-1}$ [30].

**Table 3.** Effects of tannin inclusion on rumen degradation parameters of peanut meal.

| Item [1] | Tannin Inclusion, g/kg | | | | SEM [2] | p-Value | | |
|---|---|---|---|---|---|---|---|---|
| | **0** | **20** | **40** | **60** | | **Linear** | **Quadratic** | **Cubic** |
| Dry matter | | | | | | | | |
| A, g/kg | 250 | 228 | 212 | 231 | 398 | 0.04 | <0.01 | 0.40 |
| B, g/kg | 757 | 832 | 749 | 791 | 23.5 | 0.86 | 0.48 | <0.01 |
| kd, h$^{-1}$ | 0.043 | 0.040 | 0.047 | 0.036 | 0.003 | 0.56 | 0.38 | 0.15 |
| Crude protein | | | | | | | | |
| A, g/kg | 195 | 219 | 226 | 128 | 3.50 | <0.01 | <0.01 | 0.07 |
| B, g/kg | 829 | 772 | 838 | 885 | 13.9 | 0.08 | 0.08 | 0.29 |
| kd, h$^{-1}$ | 0.041 | 0.048 | 0.037 | 0.040 | 0.001 | 0.61 | 0.72 | 0.25 |

[1] A, water-soluble fraction; B, potentially degradable water-insoluble fraction; kd, degradation rate of fraction B; [2] SEM, standard error of the mean.

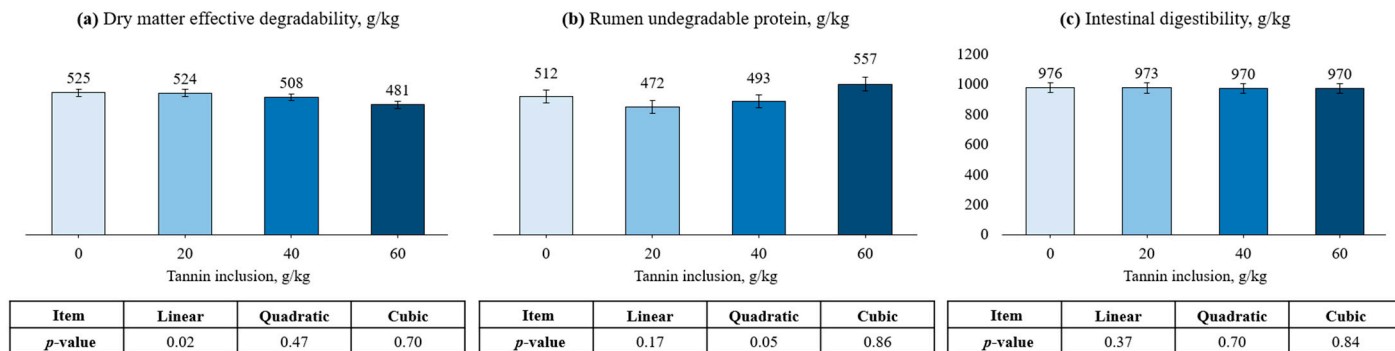

**Figure 2.** Effects of tannin treatment on dry matter effective degradability ((**a**), ED), rumen undegradable protein ((**b**), RUP), and intestinal digestibility (**c**), of peanut meal. ED = A + [B × kd/(kd + kp) × e-kt] [29]; RUP = B × [kp/(kp + kd)], where A is the water-soluble fraction, g/kg; B is the potentially degradable water-insoluble fraction, g/kg; kd is the degradation rate of fraction b, h$^{-1}$; t is time, h; and kp is the rumen passage rate of 0.074 h$^{-1}$ [30].

### 3.3. Soybean Meal

Regarding DM degradation kinetics, fractions A and B, and ED had cubic effects (*p* < 0.01), being greater for control compared with the other treatments and responded quadratically as tannin increased (Table 4 and Figure 3). However, tannin levels did not affect kd ($P_{Lin.}$ = 0.68; $P_{Quad.}$ = 0.41; $P_{Cub.}$ = 0.34). For CP kinetics, tannin inclusion had a quadratic effect on fraction kd, heating the greatest value at 60 g/kg (*p* < 0.01). Also, tannin

inclusion had a cubic effect on fractions A and B, RUP, and ID ($p < 0.01$). The cubic behavior showed greater B fraction and ID, and lower A fraction and RUP for control, compared other treatments. Then, tannin inclusion had a quadratic effect on these parameters.

**Table 4.** Effects of tannin inclusion on rumen degradation parameters of soybean meal.

| Item [1] | Tannin Inclusion, g/kg | | | | SEM [2] | *p*-Value | | |
|---|---|---|---|---|---|---|---|---|
| | 0 | 20 | 40 | 60 | | Linear | Quadratic | Cubic |
| Dry matter | | | | | | | | |
| A, g/kg | 287 | 182 | 208 | 183 | 3.19 | <0.01 | <0.01 | <0.01 |
| B, g/kg | 731 | 698 | 785 | 667 | 6.43 | 0.32 | 0.08 | <0.01 |
| kd, h$^{-1}$ | 0.049 | 0.042 | 0.049 | 0.050 | 0.002 | 0.68 | 0.41 | 0.34 |
| Crude protein | | | | | | | | |
| A, g/kg | 128 | 147 | 197 | 117 | 5.02 | 0.70 | <0.01 | <0.01 |
| B, g/kg | 912 | 811 | 926 | 903 | 17.4 | <0.01 | <0.01 | <0.01 |
| kd, h$^{-1}$ | 0.038 | 0.025 | 0.030 | 0.062 | 0.002 | <0.01 | <0.01 | 0.71 |

[1] A, water-soluble fraction; B, potentially degradable water-insoluble fraction; kd, degradation rate of fraction B; [2] SEM, standard error of the mean.

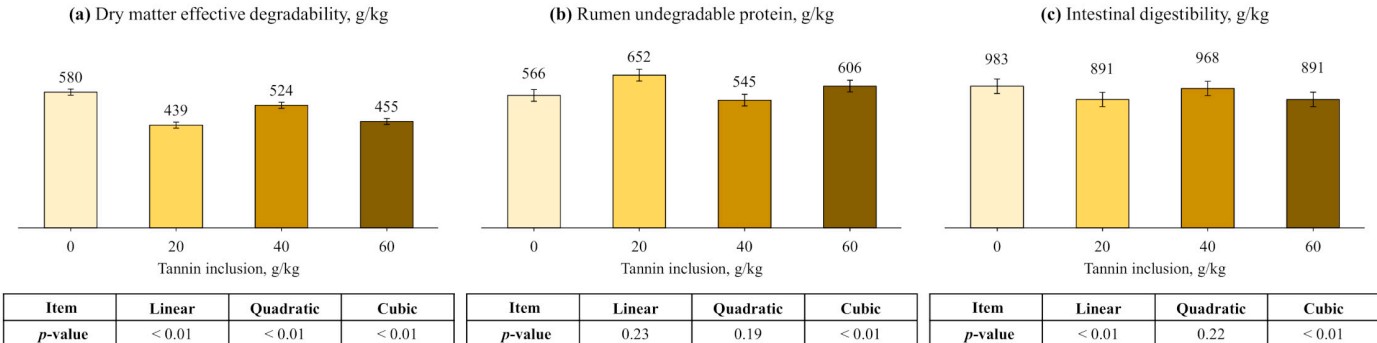

**Figure 3.** Effects of tannin treatment on dry matter effective degradability ((**a**), ED), rumen undegradable protein ((**b**), RUP), and intestinal digestibility (**c**), of soybean meal. ED = A + [B × kd/(kd + kp) × e-kt] [29]; RUP = B × [kp/(kp + kd)], where A is the water-soluble fraction, g/kg; B is the potentially degradable water-insoluble fraction, g/kg; kd is the degradation rate of fraction b, h$^{-1}$; t is time, h; and kp is the rumen passage rate of 0.074 h$^{-1}$ [30].

## 4. Discussion

The rise of the rumen undegradable protein (RUP) content of a diet is essential to fulfilling the requirements of high-performance animals [6]. In that regard, tannins can be utilized to increase RUP by their capacity to form complexes with diverse nutrients present in the feed [21,33]. However, to our knowledge, the optimal levels of tannin inclusion on protecting the amino acids of common protein sources (cottonseed, peanut, and soybean meals) from ruminal fermentation remain unclear. Regarding ruminal kinetic parameters, fraction A is soluble and readily available for ruminal degradation, fraction B is potentially degraded in the rumen within a specific timeframe, and kd is the degradation rate of fraction B [34,35]. Thus, we hypothesized that the treatment of these feed with tannin could modify the ruminal kinetic parameters of DM and CP. Indeed, tannin inclusion reduced the soluble fraction of DM for all tested feed, reaching lower values at 40 g/kg for cottonseed and peanut meals and 60 g/kg for soybean meal. Interestingly, the tannin effect was higher on protein soluble fraction, where 60 g/kg was the treatment with lower values for all feeds. The explanation for these results lies in the fact that tannin treatment can facilitate the creation of complex compounds with low solubility in the rumen [36]. Moreover, hydrogen bond interactions between tannin and protein could be the reason for the higher effect on protein soluble fraction [37,38]. The inclusion of tannins has also resulted in a reduction in the percentage of protein degradation in the rumen over time for both cottonseed and

soybean meals, at 60 and 20 g/kg, respectively. This reduction in Kd indicates a shorter duration for the protein ruminal fermentation; this, in turn, can facilitate the enhancement of amino acid flow and absorption in the animal's intestine [23,39,40]. Thus, our results suggest that tannin treatment changed ruminal nutrient kinetics by protecting the feed from ruminal fermentation.

The use of reduced concentrations of tannins in the diet can be an alternative for optimizing protein supply to ruminants [41]. This is attributed to the capacity of tannins to bind with proteins and form tannin–protein complexes. This alteration in fermentation kinetics promotes greater ruminal escape and consequently increases RUP levels [42]. Thus, the changes in the ruminal kinetics caused by tannin inclusion were reflected in the reduction in ED and an increase in RUP levels, as expected. Regarding ED, caution must be exercised when including tannins as their usage may lead to toxic effects on ruminal microorganisms [43]. This could result in a reduction in fiber degradation and organic matter digestibility [43,44]. The utilization of condensed tannins could lead to a reduction in substrate degradation by hindering the attachment of microbes to feed particles and by forming bonds with nutrients and microbial enzymes [37]. Furthermore, there is a possibility of tannin complexation with other compounds, such as starch [45]. The interaction between tannins and starch is considered pH-independent but can be influenced by factors like tannin solubility [45]. This could explain the results observed for peanut meal (a feed with higher starch content). In this case, there was a linear decrease in digestibility and a quadratic effect on RUP. Specifically, there was a reduction in digestibility for the first two tannin levels, followed by an increase in the 60 g/kg treatment. Thus, it appears that the highest tannin level offers the most favorable conditions for improving protein–tannin complexation in both cottonseed and peanut meals.

The increase in RUP implies a decrease in the quantity of substrates accessible for microbial fermentation within the ruminal environment [46,47]. However, it is important to emphasize that this protection should occur only at the ruminal level, leaving it susceptible to digestion and absorption in the other compartments of the animal's gastrointestinal tract [48]. Thus, we hypothesized that tannin treatment could increase the RUP content of the tested feeds without impairing the intestinal digestibility of protein. Indeed, the lack of differences observed for the peanut meal treatments indicates that protein–tannin complexes were sufficiently robust to shield the protein from rumen fermentation but not strong enough to prevent their breakdown after passing through the rumen. The complexes formed for feed protection between protein and tannin should be reversible when exposed to pH values lower than the ruminal pH, such as those found in the abomasum and small intestine [21,49]. On the other hand, the cubic effect observed for the protein intestinal digestibility of cottonseed and soybean meals suggests that higher levels of tannin inclusion (60 g/kg) could also have a detrimental impact on post-rumen protein utilization.

Therefore, our results lead us to the conclusion that tannin treatment may enhance the RUP content of cottonseed and peanut meals. However, caution must be exercised regarding the levels of tannin inclusion as an excess of tannins could potentially impact the availability of amino acids for intestinal absorption, as observed with soybean meal. For this feed, the alteration in ruminal kinetics has also impacted the RUP, but the reductions in intestinal digestibility did not lead to an improvement in protein utilization. In fact, the control treatment exhibited 1.01%, 2.91%, and 3.36% more RUP being digested in the intestine compared to the increasing levels of tannin inclusion. Considering that the reactivity of tannins with proteins can be influenced by factors such as the chemical structure, molecular weight of tannins, and amino acid content of proteins [40,45,50], it appears that soybean meal was less affected by these treatments. Some studies observed that doses of up to 20 g/kg of dry matter (DM) resulted in reduced protein digestibility when Jersey steers were fed diets based on soybean meal [51]. According to these authors, this effect may be attributed to several factors, including the binding effect of tannins with proteins, the lack of dissociation of a portion of the tannin–protein complex in the abomasum, potential inactivation of intestinal enzymes, or re-binding to proteins.

In summary, the amount of RUP digested in the intestine was as follows: for cottonseed meal: 0 g/kg tannin inclusion (304 g/kg), 20 g/kg tannin inclusion (375 g/kg), 40 g/kg tannin inclusion (392 g/kg), and 60 g/kg tannin inclusion (401 g/kg); for peanut meal: 0 g/kg tannin inclusion (500 g/kg), 20 g/kg tannin inclusion (459 g/kg), 40 g/kg tannin inclusion (478 g/kg), and 60 g/kg tannin inclusion (540 g/kg). Thus, our results suggest the potential inclusion of tannins for each feed. Regarding both cottonseed and peanut meals, we highlight the inclusion of 60 g/kg of tannins, which resulted in 31.9% and 8.14% more RUP being digested in the intestine compared to the non-processed treatment, respectively. Subsequent actions will involve conducting studies to assess the rumen fermentation parameters of these ingredients.

## 5. Conclusions

Tannins present a promising alternative for enhancing RUP levels in diets containing cottonseed and peanut meals. However, the effectiveness of tannins is contingent upon their dosage. Therefore, the most favorable outcomes under these experimental conditions were observed with a tannin inclusion level of 60 g/kg for both feeds. This dosage indicates effective protection of protein from ruminal microbial degradation without compromising its absorption at the intestinal level. This suggests that tannins can serve as a viable alternative for processing these feedstuffs. However, no benefits were observed from treating soybean meal with tannin. Concerning the treated cottonseed and peanut meals, it is advisable to conduct additional research to assess the impact of these ingredients on ruminal fermentation parameters and animal performance.

**Author Contributions:** Conceptualization, P.D.B.B. and E.M.P.; methodology, K.E.L., D.A.B.P., F.R., E.M., E.A.B. and E.M.P.; formal analysis, P.D.B.B., M.I.M. and E.M.P.; investigation, K.E.L., D.A.B.P., P.D.B.B. and E.M.P.; resources, R.H.B. and E.M.P.; data curation, K.E.L., D.A.B.P., M.I.M., P.D.B.B. and E.M.P.; writing—original draft preparation, K.E.L., D.A.B.P., M.I.M., P.D.B.B. and E.M.P.; writing—review and editing, K.E.L., D.A.B.P., F.R., E.M., M.I.M., R.H.B., P.D.B.B. and E.M.P.; visualization, K.E.L., D.A.B.P., F.R., E.M., M.I.M., R.H.B., P.D.B.B. and E.M.P.; supervision, R.H.B., P.D.B.B. and E.M.P.; project administration, R.H.B. and E.M.P.; funding acquisition, R.H.B., P.D.B.B. and E.M.P. All authors have read and agreed to the published version of the manuscript.

**Funding:** The authors gratefully acknowledge the São Paulo Research Foundation (FAPESP) for funding the project (Grant# 2018/19743-7; Grant # 2017/50339-5). They also thank the members of the Laboratory of Nutrition and Ruminal Fermentation of the Beef Cattle Research Center of Instituto de Zootecnia for their assistance with sample collection and laboratorial analyses. The authors also acknowledge the funding received from the Fundação de Amparo à Pesquisa e Inovação do Estado de Santa Catarina (FAPESC, Public notice number: 48/2022, grant: 2023TR000535).

**Institutional Review Board Statement:** This study was carried out in strict accordance with the recommendations of the Animal Use Ethics Committee of the Instituto de Zootecnia. Animal care and handling protocols were approved by this committee (Protocol Number: 249-19).

**Informed Consent Statement:** Not applicable.

**Data Availability Statement:** The data presented in this study are available upon request from the corresponding author.

**Acknowledgments:** The authors gratefully acknowledge the farm crew at the Experimental Station (Instituto de Zootecnia) for animal feeding and care.

**Conflicts of Interest:** The authors declare no conflict of interest.

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
