# Peer review of "Effect of Tannin Inclusion on the Enhancement of Rumen Undegradable Protein of Different Protein Sources"

_ruminants, doi:10.3390/ruminants3040034_

Round 1

Reviewer 1 Report

Comments and Suggestions for Authors

This paper reports the effects of Acacia mearnsii tannins treatment on the degradability of different protein sources. Although the study is not new, the study results would be useful.

L85-89: Please describe how the protein sources were treated with tannins.

L88: check the spelling of mearnssi. mearnsii?

L91-96: a better description is needed for the design - how many samples were incubated for each protein sources and in each animal or digestion vessel?

Figures may be removed and the data can be presented in each table for the protein sources.

L309-310: which amino acids differed in soybean than other two protein sources, which resulted in differences in protein protein degardability rates? More exploration in this regard is needed for the explanation rather than conventional explanation.

What was cumulative degradability in both ruminal and intestinal parts for each protein sources. This would be beneficial to understand overall effect of tannin treatment.

Comments on the Quality of English Language

English is readable.

Author Response

REVIEWER 1

This paper reports the effects of Acacia mearnsii tannins treatment on the degradability of different protein sources. Although the study is not new, the study results would be useful.

AU: Thank you for your time and constructive criticisms regarding our manuscript, we are confident that your thoughtful comments have substantially improved this manuscript and we sincerely appreciate them. You bring up excellent points and we have addressed your recommendations throughout the manuscript.

Please find below the answers (in red) to your comments.

L85-89: Please describe how the protein sources were treated with tannins.

AU: We added this description to address your recommendation in the manuscript.

L88: check the spelling of mearnssi. mearnsii?

AU: Text was changed to reflect your recommendation.

L91-96: a better description is needed for the design - how many samples were incubated for each protein sources and in each animal or digestion vessel?

AU: Text was changed to reflect your recommendation.

Figures may be removed and the data can be presented in each table for the protein sources.

AU: We believe that presenting the main evaluated parameters in figures is better for results presentation.

L309-310: which amino acids differed in soybean than other two protein sources, which resulted in differences in protein protein degardability rates? More exploration in this regard is needed for the explanation rather than conventional explanation.

AU: We understand your point; however, we did not evaluate the amino acid profile of the feed used in our study. Thus, including this discussion in our manuscript would be speculative. Therefore, we prefer not to include it.

What was cumulative degradability in both ruminal and intestinal parts for each protein sources. This would be beneficial to understand overall effect of tannin treatment.

AU: Thank you for the suggestion; we have included this information in the discussion section.

Reviewer 2 Report

Comments and Suggestions for Authors

Lines 35-36 - The Albino, PIRES and Rostagno referenves are not in the list of references. Add these references.

Line 106 - You refer to Table 1 but there is no table provided in the text with chemical composition of the ingredients. Add this table.

Line 106 - Remove the word presented. Tsbles and figures can't present. Make this change in the entire paper.

Line 108 - Change to Table 2. Define EPM

LIne 182 - protein ID is not in Table 2 but it is in Figue 1. CHsnge wording on this.

Line 184 - DM and CP A frctions linear effect is also signifcant. Reword to reflect this.

Line 207 - Define DE

Line 212 - Change to Table 4

Line 262 - Complete or delete the partial sentence

Line 377 - Change reference to inclide Stern

LIne 352 - Changec spelling of referenced in  this list to match the text match 

All references in both text and reference list - Make conssitent in terms of captilazation and lower case letters

Comments on the Quality of English Language

Consistency of capitilazation and lower case neds to be corrected n both the text and the reference list.

Author Response

REVIEWER 2

Lines 35-36 - The Albino, PIRES and Rostagno referenves are not in the list of references. Add these references.

AU: References were added to reflect your recommendation.

Line 106 - You refer to Table 1 but there is no table provided in the text with chemical composition of the ingredients. Add this table.

AU: Thank you. We have added Table 1.

Line 106 - Remove the word presented. Tsbles and figures can't present. Make this change in the entire paper.

AU: Text was changed to reflect your recommendation.

Line 108 - Change to Table 2. Define EPM

AU: EPM stands for SEM (standard error of the mean). We changed it.

LIne 182 - protein ID is not in Table 2 but it is in Figue 1. CHsnge wording on this.

AU: Text was changed to reflect your recommendation.

Line 184 - DM and CP A frctions linear effect is also signifcant. Reword to reflect this.

AU: Text was changed to reflect your recommendation.

Line 207 - Define DE

AU: The acronym was incorrect; the correct one is ED, which stands for Effective Degradability. We changed it.

Line 212 - Change to Table 4

AU: Text was changed to reflect your recommendation

Line 262 - Complete or delete the partial sentence

AU: Text was changed to reflect your recommendation

Line 377 - Change reference to inclide Stern

AU: Text was changed to reflect your recommendation

Line 352 - Changec spelling of referenced in this list to match the text match

AU: References were adjusted following Ruminants' citation guidelines.

All references in both text and reference list - Make conssitent in terms of captilazation and lower case letters

Consistency of capitilazation and lower case neds to be corrected n both the text and the reference list.

AU: References were adjusted following Ruminants' citation guidelines.

Reviewer 3 Report

Comments and Suggestions for Authors

The abstract is overall well written, but the authors need to include more information. Specifically, more information on the experimental design in the abstract is warranted – how many animals were used? How long was the time frame for feeding and measurements? At line 18, the authors need to include P values in the abstract. Whenever the authors discuss changes or differences in the data, the P values need to be reported in the data. This is true at line 20 as well and elsewhere throughout the abstract. The authors also need to describe what fractions A and B and the kd are in the abstract.

The introduction is overall good – there are just a few minor points that need to be addressed:

-At line 43 productive should be changed to producing.

-The first sentence of the paragraph beginning at line 45 needs to be re-worded to address location and put this into context. The word “the” also needs removed from this sentence.

-At line 74, I think the authors misspoke – wouldn’t you hypothesize that tanning increase RUP rather than RDP? Please double check this point.

The authors need to include more information in the methods. The methods and experimental design are not well described and much more detail is needed.

-How old were the bulls?

-Were the protein and tannin sources given as part of a TMR or were the protein sources themselves evaluated?

-Where did the rumen fluid come from in the fermenters?

-Did the data in table 1 come from a fermenter trial or an animal trial?

-Were the experiments for each protein source run at the same time or each individually?

-What was the experimental unit in the statistical analysis? Was it each individual bag or the animal? The n needs to be included in the statistical analysis section.

The results are overall well written, but EPM needs to be defined in the tables – is this SEM?

In the discussion, the sentence at line 241 needs to be re-worded. Optimal levels rather than better levels, perhaps? Overall though, the discussion is generally well written.

A general note is that some authors in references are all caps and some are not – this needs to be consistent throughout the paper and meet the formatting guidelines of the journal.

Comments on the Quality of English Language

Only minor editing is required

Author Response

REVIEWER 3

The abstract is overall well written, but the authors need to include more information. Specifically, more information on the experimental design in the abstract is warranted – how many animals were used? How long was the time frame for feeding and measurements? At line 18, the authors need to include P values in the abstract. Whenever the authors discuss changes or differences in the data, the P values need to be reported in the data. This is true at line 20 as well and elsewhere throughout the abstract. The authors also need to describe what fractions A and B and the kd are in the abstract.

The introduction is overall good – there are just a few minor points that need to be addressed:

AU: Thank you! We appreciate your nice and thoughtful comments and we are confident that your comments have substantially improved this manuscript and we sincerely appreciate them.

We have improved the material and methods sections to reflect your recommendations. Three animals were used for in situ incubation. Animals were adapted for 14 days prior to the incubations. In situ incubation lasted for 2, 4, 8, 16, 24, or 48 h. In vitro incubation lasted for 24h. All analyses were performed just after incubation. Also, we have added p values throughout the abstract and the description of fractions A, B, and kd.

Please find below the answers (in red) to your specific comments.

At line 43 productive should be changed to producing.

AU: Text was changed to reflect your recommendation

The first sentence of the paragraph beginning at line 45 needs to be re-worded to address location and put this into context. The word “the” also needs removed from this sentence.

AU: Text was changed to reflect your recommendation

At line 74, I think the authors misspoke – wouldn’t you hypothesize that tanning increase RUP rather than RDP? Please double check this point.

AU: Yes, you are right. Text was changed to reflect your recommendation

The authors need to include more information in the methods. The methods and experimental design are not well described and much more detail is needed.

AU: We have improved the information in the experimental design section.

How old were the bulls?

AU: 24 months old. We have added this to the manuscript.

Were the protein and tannin sources given as part of a TMR or were the protein sources themselves evaluated?

AU: First, protein sources were treated with tannin (we have added this description to the manuscript). Then, the treated protein sources were evaluated.

Where did the rumen fluid come from in the fermenters?

AU: The rumen fluid came from the same cannulated animals used in the in situ studies."

Did the data in table 1 come from a fermenter trial or an animal trial?

AU: We have corrected Table 1.

Were the experiments for each protein source run at the same time or each individually?

AU: Each of the three experiments was run individually.

What was the experimental unit in the statistical analysis? Was it each individual bag or the animal? The n needs to be included in the statistical analysis section.

AU: Bags were used as experimental units, and we have improved the information in the experimental design section.

The results are overall well written, but EPM needs to be defined in the tables – is this SEM?

AU: EPM stands for SEM (standard error of the mean). We changed it.

In the discussion, the sentence at line 241 needs to be re-worded. Optimal levels rather than better levels, perhaps? Overall though, the discussion is generally well written.

AU: Thank you! Text was changed to reflect your recommendation

A general note is that some authors in references are all caps and some are not – this needs to be consistent throughout the paper and meet the formatting guidelines of the journal.

AU: References were adjusted following Ruminants' citation guidelines.

Reviewer 4 Report

Comments and Suggestions for Authors

Overall, the data is relatively small and unsystematic, and more experimental data needs to be added to illustrate the results of this experiment. The unit in all tables should be doulble checked (g/kg).

The data in this manuscript would be more appropriate in the form of a technical report and would not be appropriate for publication as a full scientific paper.

1. Line 2, What does the acronym 'DE' stand for?

2. There were two 'Figure 2', Line 261 and line 197. pls double-check the title of the figures

3.the composition and nutritive value of the diets for the cannulated bulls were needed. pls add these data.

4.the effective degradation rate (ED) values should be presented in the tables. 

5. What does 'EPM' stand for in all the tables?

6. How to calibrate the sample escape rate ? 

Comments on the Quality of English Language

Moderate editing of English language required

Author Response

Overall, the data is relatively small and unsystematic, and more experimental data needs to be added to illustrate the results of this experiment. The unit in all tables should be doulble checked (g/kg).

AU: Thank you for your time and constructive criticisms regarding our manuscript. We have addressed your recommendations throughout the manuscript.

Please find below the answers (in red) to your comments.

The data in this manuscript would be more appropriate in the form of a technical report and would not be appropriate for publication as a full scientific paper.

AU: Thanks for the suggestion.

  1. Line 2, What does the acronym 'DE' stand for?

AU: EPM stands for SEM (standard error of the mean). We changed it.

  1. There were two 'Figure 2', Line 261 and line 197. pls double-check the title of the figures

AU: Thank you for point it out. Text was changed to reflect your recommendation

3.the composition and nutritive value of the diets for the cannulated bulls were needed. pls add these data.

AU: Text was changed to reflect your recommendation

4.the effective degradation rate (ED) values should be presented in the tables.

AU: We understand your point, but we believe that presenting the main evaluated parameters in figures is better for results presentation. Thus, we prefer to keep ED in the figures.

  1. What does 'EPM' stand for in all the tables?

AU: EPM stands for SEM (standard error of the mean). We changed it.

  1. How to calibrate the sample escape rate?

AU: We didn’t follow your question here. For both in situ and in vitro trials, samples were placed in filter bags with a limited porosity that allows microbial access and escape of fermentation products. Anyway, degradation rate (kd) was estimated according to Orskov and McDonald (1979), and used rumen passage rate (kp) was 0.074 h-1 (NRC, 2001).

NRC, Nutrient Requirements of Dairy Cattle, 7º revised edition; National Academy Press, Washington, DC, USA, 2001. ISBN 0309069971.

Orskov, E.; McDonald, I. The estimation of protein degradability in the rumen from incubation measurements weighted according to rate of passage. Journal of Agricultural Science 1979, v.92, p.499-503.

Round 2

Reviewer 3 Report

Comments and Suggestions for Authors

Thank you for addressing all of my concerns. No additional comments

Comments on the Quality of English Language

Overall, this looks good. 

Reviewer 4 Report

Comments and Suggestions for Authors

No